# Context-Dependent Ant-Pollinator Mutualism Impacts Fruit Set in a Hummingbird-Pollinated Plant

**DOI:** 10.3390/plants12213688

**Published:** 2023-10-26

**Authors:** Priscila Bruno Cardoso, Eduardo Soares Calixto, Helena Maura Torezan-Silingardi, Kleber Del-Claro

**Affiliations:** 1Postgraduation Program in Entomology, Department of Biology, University of São Paulo, Ribeirao Preto 14040-900, SP, Brazilhmtsilingardi@gmail.com (H.M.T.-S.); 2Entomology and Nematology Department, University of Florida, Jay, FL 32583, USA; 3Institute of Biology, Universidade Federal de Uberlândia, Uberlandia 38405-302, MG, Brazil

**Keywords:** *Palicourea rigida*, plant fitness, cerrado, pericarpial nectaries, extrafloral nectaries, *Camponotus crassus*, *Ectatomma tuberculatum*

## Abstract

Context-dependence in mutualisms is a fundamental aspect of ecological interactions. Within plant-ant mutualisms, particularly in terms of biotic protection and pollination, research has predominantly focused on elucidating the benefits while largely overlooking potential costs. This notable gap underscores the need for investigations into the drawbacks and trade-offs associated with such mutualistic relationships. Here, we evaluated the role of pericarpial nectaries (PNs) in shaping the dynamics of ant-pollinator mutualisms. Specifically, we investigated whether ants visiting the PN of *Palicourea rigida* (Rubiaceae) could deter hummingbirds and disrupt pollination, ultimately influencing fruit production. Our research involved manipulative experiments and observation of ant-pollinator interactions on *P. rigida* plants in the Brazilian savannah. We found that visiting ants can deter hummingbirds and/or disrupt pollination in *P. rigida*, directly influencing fruit set. However, these results are species-specific. The presence of very aggressive, large predatory ants, such as *E. tuberculatum*, had a negative impact on hummingbird behavior, whereas aggressive mid-sized ants, such as *C. crassus*, showed no effects. Our study illuminates the multifaceted aspects of ant-plant mutualisms and underscores the importance of evaluating costs and unexpected outcomes within these ecological relationships.

## 1. Introduction

Mutualistic interactions between ants and plants play a crucial role in shaping ecological communities and influencing plant fitness [1,2]. These interactions are often mediated by food resources provided by plants, such as extrafloral nectar and floral nectar [3,4,5,6]. Ants, in turn, provide various services to the plants, including protection against herbivores [7]. However, the costs and benefits of these mutualisms can vary depending on multiple factors, such as the identity of ant species, resource availability, and environmental conditions [8,9,10,11]. Several studies have investigated the trade-offs associated with these mutualisms, highlighting both positive and negative impacts on plant fitness [12,13]. For example, ants’ protective services can enhance plant survival by reducing herbivory, while their foraging activities may negatively affect mutualists [7]. Additionally, the presence of ants on plants can affect pollination dynamics by deterring pollinators [14,15,16,17,18]. Consequently, the outcomes of ant-plant mutualisms are context-dependent and influenced by intricate ecological interactions [1,19,20]. 

Extrafloral nectaries (EFNs) and pericarpial nectaries (PNs) are structures that produce sugar-rich secretions, attracting ants and facilitating their mutualistic interactions with plants [5,21,22]. EFNs are glandular structures located outside the flowers [4], while PNs are found on developing fruits [17,23]. These nectaries serve as a valuable food resource for ants [24,25], promoting their colonization and defensive activities on the plants. Ants, in turn, provide plant protection against herbivores, including leaf-chewing insects and seed predators. The presence of ants can significantly reduce herbivory levels, increasing plant fitness and reproductive success [26,27,28]. However, the benefits of ant-plant mutualisms may be compromised under certain circumstances. For instance, high ant abundance can lead to aggressive behaviors and interfere with other pollinators [29], thereby negatively impacting plant reproductive success. Moreover, in some cases, ants may engage in exploitative behaviors, consuming floral resources without providing any significant benefits to the plant [22]. Moreover, in some cases, ants may engage in exploitative behaviors, consuming floral resources without providing any significant benefits to the plant [9,22,30]. Thus, understanding the consequences of ant presence on plant fitness requires a comprehensive assessment of the costs and benefits associated with these mutualistic interactions, considering both the direct and indirect effects on plant reproductive success and community dynamics.

While PNs can attract ants, their presence on developing fruits can have unintended consequences for plant reproductive success. As some flowers transition into fruiting stages, the production of pericarpial nectar may continue, attracting ants to the developing fruits [31,32]. Within an inflorescence, flowers can be in different phenological stages, with some actively attracting ants through their pericarpial nectaries, while others may have recently opened or be in the form of flower buds (as in Figure 1d,e). The presence of ants on fruits can deter potential pollinators, including flower-visiting insects, thereby disrupting the pollination process [17]. Thus, ants may physically block access to flowers or interfere with pollinators’ movement, ultimately reducing pollination and seed set efficiency. These interactions highlight the intricate trade-offs between ant-mediated plant protection and the potential negative impacts on pollination dynamics. 

Plant species in the Brazilian tropical savanna belonging to the Rubiaceae family commonly possess PNs, such as *Palicourea rigida* [23], *Cordiera elliptica* [32], and *Tocoyena formosa* [33,34]. In *P. rigida*, the PNs attract ants that feed on the post-floral secretions and prey on or chase away chewing herbivores, significantly reducing leaf-area loss [23]. However, no positive impact on fruit set production was observed, suggesting that ants could not deter seed-parasitic wasps. Very similar results to those of *P. rigida* were also recorded in *T. formosa* [33,34]. Shrubs of *P. rigida* have odorless flowers, have brightly colored (red to yellow) tubular corolla, and are cross-pollinated dependent, primarily facilitated by hummingbirds [35]. Considering that recent studies have shown that EFN-visiting ants can deter insect pollinators, directly influencing fruit and seed production [14,15,16,17,18,36,37], we ask the following questions: Could ants be able to inhibit a visit of a flying vertebrate pollinator? Does the ant-protecting mutualism between ants and PNs-bearing plants present an undesirable cost? Our main hypothesis is that PNs-visiting ants can deter hummingbirds and disrupt pollination, directly influencing fruit production. A series of manipulative experiments were performed to test our main hypothesis (Table 1).

## 2. Results

### 2.1. Floral Visitors and Ants

The floral visitors most frequently observed were hummingbirds of the species *Amazilia fimbriata* (Gmelin, 1788; Figure 2a), *Eupetomena macroura* (Gmelin, 1788; Figure 2b), and *Chlorostilbon lucidus* (Shaw, 1812; Figure 2c). Lepidopterans of the genera *Heraclides* sp. and *Urbanus* sp. were observed collecting nectar from the flowers. A variety of bee species also visited the plant: *Epicharis cockerelli* Friese, 1900; *Euglossa* sp.; *Ceratina* sp. 1; *Ceratina* sp. 2; *Xylocopa* sp.; *Oxaea flavescens* Klug, 1807; *Augochloropsis* sp.; *Trigona spinipes* Fabricius, 1793; and *Apis mellifera* Linnaeus, 1758 (Table 2). The ant species observed on the plant were: *Camponotus crassus* Mayr, 1862 (Figure 1e,f); *Crematogaster* sp., *Brachymyrmex* sp., *Dolichoderus* sp., *Ectatomma tuberculatum* (Oliver, 1792) (Figure 1d), and *Pseudomyrmex gracillis* (Fabricius, 1804).

### 2.2. H1: Ants Act as Protective Mutualists 

#### 2.2.1. Experiment 1

We did not find any effect of leaf age (Wald χ^2^ = 2.7, df = 1, *p* = 0.10), treatment (Wald χ^2^ = 0.34, df = 1, *p* = 0.55), and the interaction between leaf age and treatment on the herbivory rate (Wald χ^2^ = 0.04, df = 1, *p* = 0.84) (Figure 2a). Overall, the results suggest that neither the treatment factor nor the age factor had a significant impact on the herbivory response, and there was no significant interaction between the two factors. Newly flushed leaves suffered 8.9 ± 1.9% (mean ± SE) and 8.4 ± 1.7% damage in Control and No-ants treatment, respectively (Figure 1a). Fully developed leaves suffered 12 ± 2.1% and 10.7 ± 1.9% damage in Control and No-ants treatment, respectively (Figure 2a). 

#### 2.2.2. Experiment 2

There was a significant difference in the time required for ants to find termites, considering the interaction between the presence of PNs and the different locations within the plants (Wald χ^2^ = 76.52; *p* < 0.0001; Figure 2b). Approximately 50% of the termites located in the inflorescences of *P. rigida* were found in less than 300 s, indicating a relatively faster discovery rate than the other groups (z = 4.7, CI = 2.7–11). In contrast, approximately 50% of the termites located on leaves took around 600 s to be found by ants (z = 1.5, CI = 0.8–4). For *M. albicans*, ants found less than 50% of the termites on leaves (z = 0.01, CI = 1–1), and no termites were found on the inflorescences (Figure 2b).

### 2.3. H2: Ants Impact on Plant Reproductive Success

#### 2.3.1. Experiment 3

The percentage of fruits produced was significantly different among treatments (Wald χ^2^ = 11.2; df = 3; *p* = 0.0106; Figure 3a). Control (41 ± 7%, mean ± SE), the treatment with access to all ants, No-ants (38 ± 7%), and Plastic (41 ± 7%) treatments showed a similar percentage of fruits produced, which was 1.97, 1.82, and 1.97 times, respectively, greater than the percentage of fruits produced by the Ant treatment (20.8 ± 5%; Figure 3a), the treatment with *E. tuberculatum* ants. 

In addition, our results showed that out of the 15 plants in the Control group selected for observations of ant-pollinator interactions, 12 were dominated by *C. crassus*. The remaining three plants were dominated by *E. tuberculatum*. When the floral visitors were insects, *C. crassus* individuals showed variable behavior depending on the species (Table 3). However, when hummingbirds visited the plants, all *C. crassus* individuals avoided contact, often quickly leaving the inflorescences (PBC and ESC pers. obs.). Although the number of interactions between *E. tuberculatum* and floral visitors was low (eight), individuals of this ant species attacked floral visitors, including hummingbirds, in all occurrences. 

#### 2.3.2. Experiment 4 

There was no variation in the number of fruits produced per flower bud between treatments (Wald χ^2^ = 0.07; *p* = 0.79; Figure 3b). Control plants produced 0.10 ± 0.03 fruits per bud per plant, and plants without ants produced 0.11 ± 0.03 fruits per bud per plant (Figure 3b).

## 3. Discussion

Ants that visit plants in search of nectar, whether floral (pericarpial) or extrafloral, can commonly benefit the plants by reducing herbivory on leaves, buds, flowers, or fruits, thereby positively impacting the final fruit production [2,5,27,38]; this represents a strong benefit to the plants [28]; however, hidden costs of these mutualistic interactions may arise [19,39]. Confirming the initial observations of Taylor [40] and Machado et al. [35], our field observation showed that *P. rigida* is pollinated by hummingbirds (Figure 1, Table 2). Our results related to the system mediated by this pericarpial-nectaried plant confirm the hypothesis that in plants with PNs, visiting ants can deter hummingbirds and/or disrupt pollination, directly influencing plant fitness. However, it is important to note that these results are species-specific. The presence of very aggressive, large predatory ants, such as *E. tuberculatum*, had a negative impact on hummingbird behavior (Table 3), resulting in decreased fruit production (Figure 3), whereas aggressive mid-sized ants, such as *C. crassus*, showed no effects (Figure 3, Table 3). These findings emphasize the context-specific nature of ant-plant interactions, where both costs and benefits influence outcomes.

In the case of *P. rigida*, ants did not have a significant positive effect on reducing leaf area loss. Given the low levels of foliar herbivory, less than 9% in developing leaves and less than 12% in mature leaves, we suspect that this plant species possesses additional leaf defenses, possibly involving silicon or tannin deposition, as indicated by the hardness of the leaves. Similar results were obtained by previous authors [23,33]. Ant predation on termites used as baits in EFN-bearing plants indicates ants’ capabilities to deter or disrupt herbivores from plants [11,41,42]. Our results showed that termites were removed more rapidly and in significantly greater proportions when placed in the inflorescences and leaves of *P. rigida* than on the leaves or inflorescences of a nearby plant lacking plant biotic defenses such as EFNs or PNs. Aggressive ants, such as *C. crassus* and *E. tuberculatum*, were highly active in collecting nectar from the PNs of *P. rigida*. The former is a common ant species found in association with cerrado plants and is known for efficiently removing or chasing herbivores away from these plants [11,43,44]. According to Lange et al. [44], *C. crassus* is primarily active during the daytime, both in the rainy and dry seasons. They predominantly collect extrafloral nectar and honeydew, constituting 83.33% of their resources in the rainy period and 30% in the dry period. *Camponotus crassus* is a dominant species, especially within the vegetation at our study site in the Brazilian tropical savannah, as confirmed by Costa-Silva et al. [45]. *Ectatomma tuberculatum*, on the other hand, is also recognized as an abundant and highly aggressive ant species associated with biotic defenses in the Brazilian Cerrado savannah [15,46]. This ant is conspicuous within the vegetation [47] and is known for its aggressiveness and hierarchical dominance [48,49]. 

The presence of aggressive ants, such as *C. crassus* and *E. tuberculatum*, on the inflorescences of *P. rigida* in this region, led to avoidance behavior by hummingbirds, resulting in a significant negative impact on fruit set production, a genuine cost of this mutualism (Figure 3, Table 3). Assunção et al. [15] conducted experiments under natural conditions to test the hypothesis that the presence of ants on flowers of EFN-bearing plants might be perceived as a threat by pollinators (such as bees) and negatively affect plant fitness in terms of fruit set. Their findings revealed that ant bodyguard species, feeding on the extrafloral nectaries of Malpighiaceae *Heteropterys pteropetala*, did indeed induce avoidance behavior in pollinators. In a similar study, Nogueira et al. [18] demonstrated that the mere presence of artificial ants on flowers could adversely affect plant pollination rates. These effects included a reduction in visitation frequency, increased hesitation among pollinators, decreased time spent on the flower, and ultimately, lower fruiting rates (also see [17]). These outcomes can significantly impact a plant’s competitive advantage, which is crucial in an environment undergoing rapid degradation, such as the Cerrado. Our study highlights the profound influence that different cues (e.g., size, odor) from predatory ants can significantly shape plant-pollinator interactions. In our research, we have demonstrated, for the first time, these effects on vertebrate pollinators, specifically hummingbirds, with far-reaching consequences for overall plant fitness. 

The context-dependent nature of mutualism is illustrated in our study. When it comes to *C. crassus*, the control groups, which were predominantly dominated by this ant species (12 out of 15 plants), showed no apparent negative effects on the fruit set (Figure 3a,b). This facet of mutualism appeared to be harmonious and mutually beneficial for both the plant and the ant [25]. However, the scenario changed significantly when we introduced *E. tuberculatum* to the inflorescences (Ant treatment in Figure 3a). In this altered context, we observed a substantial reduction in fruit set, indicating that the presence of *E. tuberculatum* shifted the mutualistic relationship to one with potentially negative consequences for the plant’s reproductive success [42,50]. Furthermore, these conclusions are supported by the behavior of these two ant species toward hummingbirds. *Camponotus crassus* individuals exhibited a consistent avoidance response when hummingbirds visited (Table 3), clearly indicating a preference to avoid contact with these vertebrate pollinators. In contrast, *E. tuberculatum* displayed more aggressive and indifferent behavior, sometimes even attacking hummingbird visitors (Table 3). This striking contrast in ant behavior highlights how the same mutualistic relationship can vary dramatically based on the specific context and the species involved [39,42]. It emphasizes the importance of considering the broader ecological context when studying mutualisms and underscores their dynamic and context-dependent nature [19,20].

In plant-animal interactions, particularly in studies examining beneficial mutualisms like biotic protection, pollination, and diaspore dispersal, most research tends to prioritize proving and testing the benefits. However, it is essential to remember that all systems come with associated costs, and in some cases, unforeseen costs can alter the nature of the relationship from positive to negative or neutral [2,22]. As proposed by Bronstein [1,2], mutualistic systems are frequently exploited by organisms that reap the benefits mutualists offer without providing any benefits in return. While the natural history of these exploiters is well-documented, relatively little effort has been dedicated to analyzing their ecological or evolutionary significance within mutualism. To gain a comprehensive understanding of mutualism and its role in biodiversity, studies like ours that investigate costs and unexpected outcomes should not be underestimated.

## 4. Materials and Methods

### 4.1. The Study Site and Species

We conducted fieldwork between November 2016 and February 2017, the rainy season, in the Cerrado sensu stricto [51] of the Ecological Reserve of Clube de Caça e Pesca Itororó de Uberlândia (18°59′ S and 48°18′ W, WGS84 Datum, ~640 ha), Minas Gerais (MG) state, Brazil. The vegetation of the reserve is composed of several savannah physiognomies, with trees not taller than 8 m [52]. The mean monthly rainfall ranges between 0 and 360 mm, and the mean monthly temperature is between 20.0 and 25.5 °C, with a dry season between May and September and a rainy season between October and April [53].

The Rubiaceae *P. rigida* Kunth. is a common shrub (0.30–1.5 m tall) in the Cerrado distinguished by its yellow-red tube-shaped corollas, which exhibit adaptations for hummingbird pollination [15,40]; Figure 1. In our region, plants are usually in bloom from September to March [23]. The fleshy fruits become purple and ornithochorous throughout their development [54]. In *P. rigida*, after the corolla falls, the sepal ring remains active and produces nectar over the fruits throughout their development, which attracts ants [23]. 

### 4.2. Survey 1—Floral and PN Visitors

To know floral and PN visitors of *P. rigida*, we tagged 15 plants with very similar architecture (1 m tall, 3–5 inflorescences), distant at least 10m from each other. Every week, on sunny days, we conducted ten-minute direct observations at each plant between 6:00 a.m. and 6:00 p.m., with a ten-minute rest interval at the end of each series. Floral and PN visitors (mainly hummingbirds) were recorded photographically and collected for identification at the Behavioral Ecology and Interactions Laboratory (LECI) at the Federal University of Uberlandia (UFU). We consider effective pollinators (EFPs) as animals that legitimately harvested nectar by opening the corolla and effectively making contact with the floral structures. Floral visitors who did not pollinate the plants by accessing nectar from the base of the corolla or directly from the pericarpial nectaries without making contact with the plant’s reproductive structures were classified as visitors’ non-pollinators (VPNs).

### 4.3. H1: Ants Act as Protective Mutualists 

#### 4.3.1. Experiment 1

To experimentally test whether ants protect the plant against herbivory, we tagged 30 shrubs of ~1 m tall. Each pair of very similar plants was divided into a treatment or control group by flipping a coin. Plants of the treatment group had all ants manually excluded on the first day of the experiment and received on the basis of trunk (20 cm above soil level) a strip of adhesive paper (2 cm wide) covered by a resin that acts as a barrier against ant access (Tanglefoot^®^). Any structures, such as grasses, that could serve as bridges for ants to access the plants were removed. The control group also received on the basis of the trunk (20 cm above soil level) a strip of adhesive paper (2 cm large) covered by the resin; however, it covered only one side of the trunk to permit ants access to the plants. In both groups, we tagged three leaves in each plant in the initial phase of expansion to follow and record leaf area loss of along leaf ontogeny, that is, when flushing and when fully developed. Herbivory was calculated using digital images analyzed in the program ImageJ version 1.53, as performed by Calixto et al. [55]. 

#### 4.3.2. Experiment 2

We conducted an experimental manipulation test of predation to test whether the extrafloral nectaries (PNs) of P. rigida indeed attract ants capable of removing herbivores from the plant. We selected 15 *P. rigida* plants with a neighboring shrub (maximum 2 m away) of the same size (1 m tall) belonging to the *Miconia albicans* species (Melastomataceae), which lacks PNs, EFNs, or associated trophobiont herbivores. Next, we attached four live worker termites (*Nasutitermes* sp.) to each plant using white non-toxic glue: two on the inflorescences and two on the nearest leaf to the inflorescence. We then observed the interactions, behaviors, and whether ants removed each termite for a duration of 15 min.

### 4.4. H2: Ants Impact on Plant Reproductive Success

#### 4.4.1. Experiment 3

To investigate the potential influence of ants on the behavior of the main pollinators of *P. rigida*, i.e., hummingbirds, we conducted an experiment involving 60 similar plant individuals (~1 m tall), all of which presented inflorescences. Plants had the tallest inflorescence tagged and were separated into four groups of 15 plants each, namely, Control, Plastic, Ants, and No-ants. In the Control group, a strip of adhesive paper (2 cm wide) coated with resin (Tanglefoot^®^) was applied to the base of the trunk (20 cm above soil level). However, only one side of the trunk was covered to allow ants access to the inflorescences. In the Plastic group, all ants were manually excluded, and a strip of adhesive paper covered with resin was applied to the base of the trunk to prevent ant access completely. Any structures that could serve as bridges for ants to reach the plants, such as grasses, were removed. Three small circles of brown rubber (E.V.A—5 mm in diameter) were pinned to the inflorescences next to open flowers (Figure 1c). In the Ant group, the same treatment as the Plastic group was applied, except that three dead workers of the ant species Ectatomma tuberculatum were pinned to the inflorescence instead of plastic circles. These ants were collected from a neighboring plant. They were used because they are very common in *P. rigida* (pers. obs. PC and ESC; Figure 1d) and are highly aggressive toward any invertebrate or vertebrate present in the plants [15,50]. The ants were mounted in their natural foraging posture. Finally, in the No-ants group, we prevented ant access by using the same procedure as described above using the Tanglefoot resin.

Each plant in every group was individually observed for a duration of 30 min, ranging from 06:00 a.m. to 06:00 p.m., with a five-minute rest interval before moving on to the next plant. Within each inflorescence, we tagged ten flowers and recorded the visits made by potential pollinators. In the Control group, we specifically observed the interactions between ants and pollinators and recorded their behaviors. For the behavioral analyses, we considered an ant: (a) attack—ants that moved aggressively towards the floral visitor or assumed an attacking posture with the gaster facing forward (Figure 1e,f); (b) avoidance—ants that actively avoided contact, either by moving away or not approaching the inflorescence stem; (c) indifferent—ants that exhibited behavior other than attacking or avoiding, e.g., keep foraging.

The inflorescences of the four groups with floral buds (pre-anthesis) were bagged with bag voil on the day before experiments, and visits were allowed only during the experimental time. After the 30 minutes of analyses, the inflorescences were bagged again until fruit production. 

#### 4.4.2. Experiment 4

We looked for fruit set production to evaluate its impact on plant reproductive success. Thus, we tagged 26 other individuals of *P. rigida* and divided them into two groups (control and treatment) by the flip of a coin. Then, they received the same process described in Experiment 1 (with and without ants). Weekly, each one of the 13 shrubs of the control or treatment group was checked, and the number of buds, flowers, and fruits produced in the central and tallest inflorescence of the plant was counted.

#### 4.4.3. Data analyses

##### H1: Ants Act as Protective Mutualists

All analyses were conducted in R version 4.2.2 [56]. We fit and check residuals using the packages ‘glmmTMB’ [57] and ‘Dharma’ [58], respectively. The models’ significance was assessed using the Anova function from package ‘car’ [59]. Pairwise comparisons were assessed via estimated marginal means with the ‘emmeans’ package [60]. Survival analyses were conducted with packages ‘survival’ [61] and ‘survminer’ [62].


*Experiment 1*


To analyze the herbivory rate between leaf age (newly flushing leaf and fully developed leaf) and treatment groups (Control and Treatment), we fit a model with the herbivory rate as the response variable (Beta distribution) and the interaction between leaf age and treatment as the predictor variable. We also added plant ID as a random effect to control the spatial dependence of the data. 


*Experiment 2*


The survival model was fitted using the ‘coxph’ function from the “survival” package, where we used the function ‘Surv’ to fit the time and predation of termites. The interaction between plant species and the termite location within the plants was fit as a predictor variable. 

##### H2: Ants Impact on Plant Reproductive Success


*Experiment 3*


To compare the probability of fruit production between treatments, we fit a model with fruit production as the response variable (binomial distribution), treatments as the predictor variable, and plant ID as a random factor to account for the spatial dependence of the data. As we recorded the number of flower buds before counting the number of fruits, this analysis offers the probability of fruit production (whether produced or not) based on the count of flower buds. Pairwise comparisons were conducted with estimated marginal means. Data was back-transformed before plotting.


*Experiment 4*


To check if ant presence can influence fruit production, we calculated the fruit ratio of each plant in each group, using the number of fruits produced divided by the number of buds produced [63]. We fit a model using the fruit ratio as the response variable (Beta distribution), treatment as the predictor variable, and plant ID as the random effect.

## Figures and Tables

**Figure 1 plants-12-03688-f001:**
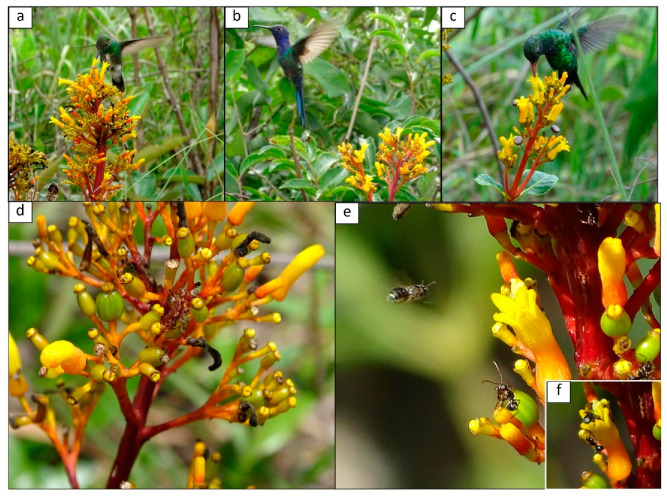
Visiting hummingbirds and ants of *Palicourea rigida*: (**a**) *Amazilia fimbriata*, (**b**) *Eupetomena macroura*, (**c**) *Chlorostilbon lucidus*, (**d**) *Ectatomma tuberculatum*, and (**e**,**f**) *Ceratina* sp. (floral visitor) and *Camponotus crassus* (ant).

**Figure 2 plants-12-03688-f002:**
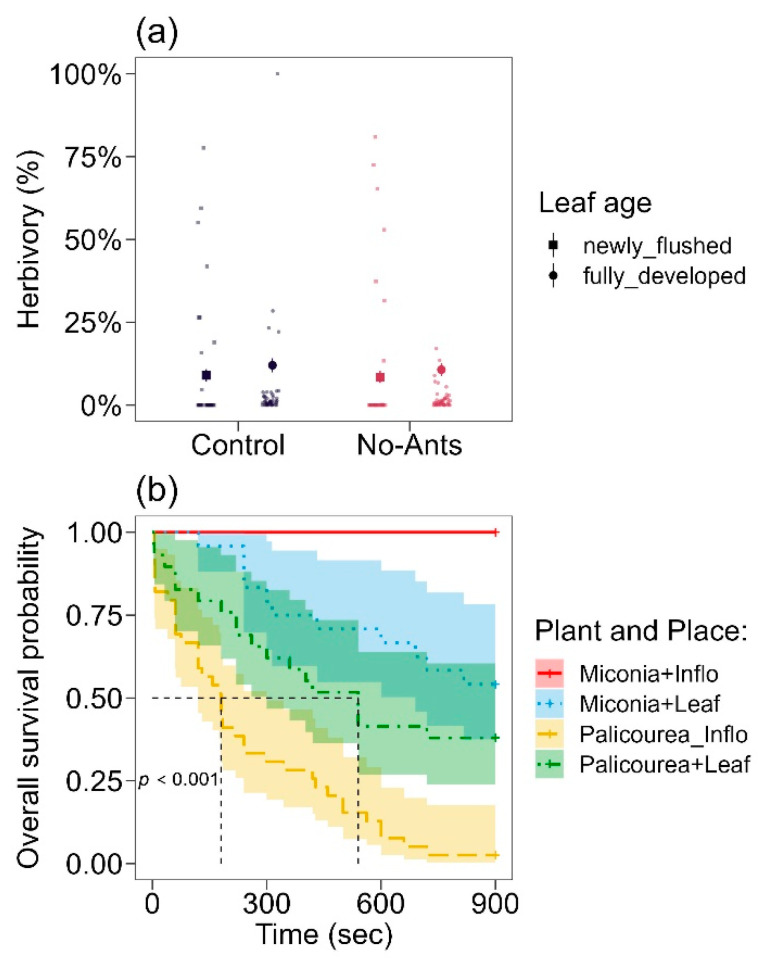
Ants effect on plant protection. (**a**) Herbivory rate on newly flushed leaves and fully developed leaves in *P. rigida*, (**b**) and survival analysis of ant predation on termites in different plant species (*P. rigida* and *M. albicans*) and locations within each plant (leaves and inflorescences). Large symbols in (**a**) represent mean ± SE (standard error), and small symbols represent raw data. Vertical and horizontal black lines in (**b**) represent the time when 50% of the termites survived in each group (more than 50% of termites survived in *Miconia*).

**Figure 3 plants-12-03688-f003:**
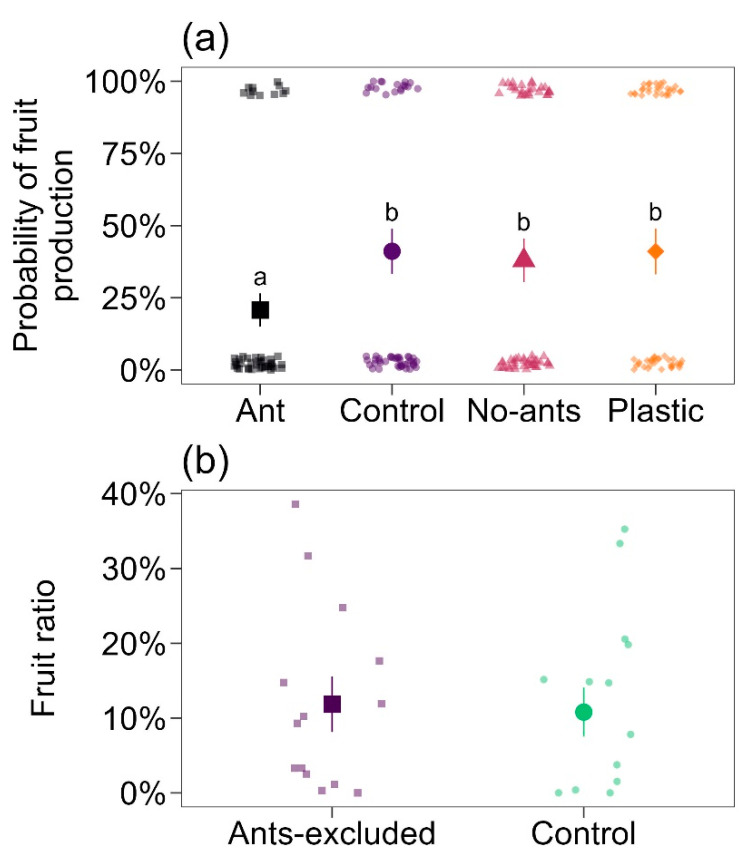
Impact of ants on fruit formation in *Palicourea rigida*. (**a**) Variation in the probability of fruit production and (**b**) fruit ratio (fruits per buds) among treatments. Large symbols represent mean ± SE, and small symbols represent raw data. Different letters in (**a**) represent significant differences based on estimated marginal means.

**Table 1 plants-12-03688-t001:** Overview of hypotheses (H) and predictions verified in this study. ^1^ Termites were used as prey, ee methodology.

Overview	Prediction	Approach	Resource
Survey 1—Floral and PN visitors	Hummingbirds are the main floral visitors, and *Camponotus crassus* and *Ectatomma tuberculatum* are the main PN visitors.	Field observations in plants without experimental manipulation.	Figure 1, Table 2
H1: Ants act as protective mutualists			
Experiment 1	A: Ants protect plants against herbivores, decreasing herbivory rate.	Evaluation of herbivory rate with and without ants.	Figure 2a
	B: Ants have more impact on the first leaf stages.	Evaluation of herbivory rate throughout leaf development with and without ants.	Figure 2a
Experiment 2	A: Ants remove herbivores faster in PN-bearing plants than in Non-PN-bearing plants.	Analysis of prey ^1^ predation.	Figure 2b
	B: Ants remove herbivores faster near PNs.	Analysis of prey ^1^ predation.	Figure 2b
H2: Ants impact on plant reproductive success			
Experiment 3	Ants on or close to flowers can repel floral visitors, including vertebrates.	Observation of floral visitors after experimental manipulation.	Figure 3a
Experiment 4	Ants can decrease plant reproductive success.	Analysis of fruit set after ant removal.	Figure 3b

**Table 2 plants-12-03688-t002:** Floral visitors in *Palicourea rigida* in a cerrado reserve in Uberlândia, MG, Brazil. ^1^ Numbers represent absolute frequency (relative frequency %). ^2^ EFP—effective pollinators, VNP—visitor non-pollinator (see methods).

Floral Visitors	AF (RF%) ^1^	Activity ^2^
**Hummingbirds**		
*Amazilia fimbriata* (Gmelin, 1788)	202 (33.6)	EFP
*Chlorostilbon lucidus* (Shaw, 1812)	253 (42.1)	EFP
*Eupetomena macroura* (Gmelin, 1788)	41 (6.83)	EFP
**Insects**		
**Hymenoptera**		
*Apis mellifera* Linnaeus, 1758	2 (0.3)	VNP
*Augochloropsis* sp.	2 (0.3)	VNP
*Ceratina* sp. ^1^	3 (0.5)	VNP
*Ceratina* sp. ^2^	12 (2.0)	VNP
*Epicharis cockerelli* Friese, 1900	6 (1.0)	VNP
*Euglossa* sp.	41 (6.8)	VNP
*Oxaea flavescens* (Klug, 1807)	15 (2.5)	VNP
*Trigona spinipes* (Fabricius, 1793)	2 (0.3)	VNP
*Xylocopa varipuncta* Patton, 1879	5 (0.8)	VNP
**Lepidoptera**		
*Heraclides* sp.	6 (1.0)	VNP
*Urbanus* sp.	10 (1.6)	VNP

**Table 3 plants-12-03688-t003:** Behavioral interactions between *Camponotus crassus* and *Ectatomma tuberculatum* against floral visitors in *Palicourea rigida* (Rubiaceae) in Cerrado, Brazil. Attack (At)—ant ran towards the floral visitor or was in attack position with the gaster facing forward (Figure 1e,f); Avoidance (Av)—avoided the contact going or not towards the inflorescence stem; Indifferent (In)—Any behavior different of attack and avoidance.

Floral Visitors	Ant Behavior					
	*Camponotus* *crassus*			*Ectatomma tuberculatum*		
	At	Av	In	At	Av	In
**Hummingbirds**						
*Amazilia fimbriata* (Gmelin, 1788)	1	11	0	2	0	0
*Chlorostilbon lucidus* (Shaw,1812)	0	13	2	0	0	0
*Eupetomena macroura* (Gmelin, 1788)	0	10	0	0	0	0
**Insects**						
**Hymenoptera**						
*Augochloropsis sp.*	1	0	1	0	0	0
*Apis mellifera* L.	0	2	0	0	0	0
*Ceratina* sp. 1	3	0	0	0	0	0
*Ceratina* sp. 2	0	0	1	0	0	0
*Epicharis cockerelli* Friese, 1900	4	1	0	1	0	0
*Euglossa* sp.	11	3	1	3	0	0
*Oxaea flavescens* (Klug, 1807)	1	3	0	0	0	0
*Polybia occidentalis* (Olivier, 1791)	1	0	0	1	0	0
*Trigona spinipes* (Fabricius, 1793)	0	0	1	0	0	0
*Xylocopa varipuncta* Patton, 1879	4	1	0	0	0	0
**Diptera**						
*Curtonotum* sp.	2	0	0	1	0	0
**Lepidoptera**						
*Heraclides* sp.	2	0	0	0	0	0
*Urbanus* sp.	3	0	1	0	0	0
**Total**	31	44	7	8	0	0

## Data Availability

The data supporting this study’s findings will be available in Dryad after manuscript acceptance.

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
