# Peer review of "Context-Dependent Ant-Pollinator Mutualism Impacts Fruit Set in a Hummingbird-Pollinated Plant"

_plants, 2023, doi:10.3390/plants12213688_

Round 1

Reviewer 1 Report

This paper examines the possible effects of ants in the fruit setting / bird pollinators well as attack by herbivores of the plant Palicourea rigida. It is a topic of interest, however, it needs some minor revisions in order to be considered for publication.Hereby are given some comments that can help the authors to produce an improved version of the manuscript.

2. Results

2.2.1. Experiment 1. If you use ANOVA as stated in the section 4.4.3. why the statistic is chi-square? Please explain. Also in the Fig 2a  it is not clear what is depicted. Error bars are not shown (probably because of the size of the fonts of the treatments ??). Please remake the fig (probably with other colors in another program), so as to be clear what is depicted.

2.3.1. Experiment 3. The "Ant" treatment is presented as % percentage (20.8%). Please correct it so as to be in line with the other treatments (0.208±0.05)

3. Discussion. 

Lines 183-184.As regards the table 3, it deems that the "visits" of Ectatoma uberculatum on the plants were scarce and lower than that of Camponotus crassus (Table 3). In the Experiment 3 C. crassus was not used as treatment. What supports that "The presence of very aggressive, large predatory ants, such as E. tuberculatum, had a negative impact on hummingbird behavior (Table 3), resulting in decreased fruit production (Fig. 3), whereas aggressive mid-sized ants, suchas C. crassus, showed no effects (Fig. 3, Table 3)." This may needs a more robust explanation. 

4. Materials and methods

The "Materials and Methods" section it is preferable to be placed before of the "Results" chapter. It is important for the reader to see and understand the experimental protocol, before seeing the results.

4.4.3. Data analysis. In this section are made some quotes (e.g Brooks et al. 2017 etc), however,  the respective references are not given in the "Reference" section. Please provide in the section "References" all the references that quoted in the text. Please keep the same type of quotation of the references throughout the manuscript, since two types of quotation are employed, i.e. numbers and the names of authors (e.g. [1], Brooks et al. 2017).

Author Response

Reviewer’s comments are followed by a Response # plus number for reference.

Reviewer: 1

This paper examines the possible effects of ants in the fruit setting / bird pollinators well as attack by herbivores of the plant Palicourea rigida. It is a topic of interest, however, it needs some minor revisions in order to be considered for publication. Hereby are given some comments that can help the authors to produce an improved version of the manuscript.

Response #1 – Thank you for your positive words and for your comments and suggestions, which we have responded to below.

  1. Results

2.2.1. Experiment 1. If you use ANOVA as stated in the section 4.4.3. why the statistic is chi-square? Please explain. Also in the Fig 2a  it is not clear what is depicted. Error bars are not shown (probably because of the size of the fonts of the treatments ??). Please remake the fig (probably with other colors in another program), so as to be clear what is depicted.

Response #2 – We used the “Anova” function (only first letter capitalized) from “car” package in R which is different from common ANOVA (Analysis of variance). This function in R determines if explanatory variables in the model are significant by using the Likelihood ratio test (if glm) or Wald chi-square test (if glmm). In our case, we used the glmm and therefore Wald chi-square test. Although the name and symbol of the test has the word “chi-square”, it is different from the chi-square test used for contingence table. In our results, we now added the word “Wald” before the symbol, so readers know we are talking about the Wald chi-square test.

We made our figure. We changed the color and decreased the size of the symbols in the plot, so that readers can now see the error bars.

2.3.1. Experiment 3. The "Ant" treatment is presented as % percentage (20.8%). Please correct it so as to be in line with the other treatments (0.208±0.05)

Response #3 – Done

  1. Discussion.

Lines 183-184.As regards the table 3, it deems that the "visits" of Ectatoma uberculatum on the plants were scarce and lower than that of Camponotus crassus (Table 3). In the Experiment 3 C. crassus was not used as treatment. What supports that "The presence of very aggressive, large predatory ants, such as E. tuberculatum, had a negative impact on hummingbird behavior (Table 3), resulting in decreased fruit production (Fig. 3), whereas aggressive mid-sized ants, suchas C. crassus, showed no effects (Fig. 3, Table 3)." This may needs a more robust explanation.

Response #4 – A more robust explanation was presented in the lines 220-223 and 239-253. Basically, this statement is based on the behavior exhibited by the ants and pollinators, the number of fruits produced in the treatments, and the number of plants dominated by each of the ants. Ectatomma tuberculatum attacked every time it interacted, while C. crassus attacked other pollinators, but rarely the main pollinators. Considering that the majority of plants in Experiment 3 (Fig. 3a) were dominated by C. crassus and the fruit production was higher than in plants with E. tuberculatum, our results suggest the potential negative effects of E. tuberculatum on pollinators when compared with crassus.

  1. Materials and methods

The "Materials and Methods" section it is preferable to be placed before of the "Results" chapter. It is important for the reader to see and understand the experimental protocol, before seeing the results.

Response #5 – We thank you for your comment. However, this sequence of sections is recommended by the Plants mdpi journal.

4.4.3. Data analysis. In this section are made some quotes (e.g Brooks et al. 2017 etc), however,  the respective references are not given in the "Reference" section. Please provide in the section "References" all the references that quoted in the text. Please keep the same type of quotation of the references throughout the manuscript, since two types of quotation are employed, i.e. numbers and the names of authors (e.g. [1], Brooks et al. 2017).

Response #6 – Done.

Reviewer 2 Report

1. In the introduction, I noticed a lack of information about the floral biology of Palicourea rigida, in particular the flowering time, and pollination requirements.

2. Line 64-69 I suggest specifying whether the statements in this section are the authors' hypothesis or citations of other authors. I also cannot agree with the statement in lines 64-65. How do ants, which are attracted by paricarpial nectaries to the fruits, disturb pollination if flower pollination occurred early before fruit set and nectar activation?

Results 

3. The title of table 2 is imprecise. The authors call all birds and insects floral visitors, and divide them into two groups EEP and and FV - floral visors using the same name.

4. Figure 2a. The labels of the control and ant group in the legend are unnecessary, because this information is presented below the X axis. The standard error is presented graphically with whiskers usually. What did the authors mean when they wrote “Large symbols represent mean±SE”?

5. If I read the legend to Figure 2b correctly, the results do not correspond to the interpretation in subsection 2.2.2. According to Fig. 2b, there are less than 50% on Miconia inflorescences after 300 seconds and on Pollicourea inflorescences after 600 sec. More than 50% of termites stay on Policourea leaves and 100% of termites are present on Miconia leaves at all times. This mistake is also in line 127. 

6. I suggest writing, the percentage of ants remaining as a percentage that survived. 

It avoids contradiction when the authors write that less than 50% termites was found by ants and the survival curve presents more than 50% 

7. I suggest unifying (percentage or decimal value) the presentation of fruit production probability and fruit ratio in the text and in Figure 3.  

8. It should be very clearly highlighted that in Experiment 3, the Ants group refers to one ant species and the Control group to other ant species. 

9. If the no-ants group and the ant-excluded group (experiment 1 and 4) are similarly constituted, why the different names?

Methods

10. Line 270-275 This paragraph does not relate directly to the study method and should rather be mentioned in the introduction.

11. How many replications of flower observations were made during November-February? Have the plants been blowing flowers all this time? 

12. What were the rules for dividing floral visitors into EEP and FV? The frequency of Euglossa bee visits was the same as E. macroura. Why is this bee excluded as a pollinator? Did insects collect nectar from pericarpial nectaries?

13. It is necessary to explain how the probability of fruit production was calculated. What was the meaning of recording and measuring the visitation time of pollinators? And were these variables included in the model of probability of the fruit  production?

Author Response

Reviewer’s comments are followed by a Response # plus number for reference.

Reviewer: 2

  1. In the introduction, I noticed a lack of information about the floral biology of Palicourea rigida, in particular the flowering time, and pollination requirements.

Response #7 – Although we have focused on describing the species in the materials and methods, we now added more information in the introduction (L. 79-80).

  1. Line 64-69 I suggest specifying whether the statements in this section are the authors' hypothesis or citations of other authors. I also cannot agree with the statement in lines 64-65. How do ants, which are attracted by paricarpial nectaries to the fruits, disturb pollination if flower pollination occurred early before fruit set and nectar activation?

Response #8 – We now made the statement clear by adding a citation when needed. “However, this presence of ants on fruits can deter potential pollinators, including flower-visiting insects, thereby disrupting the pollination process [17]. Thus, ants may physically block access to flowers or interfere with the movement of pollinators, ultimately reducing the efficiency of pollination and seed set”.

We now made clear how ants attracted to pericarpial nectaries disturb pollination. Flowers in the inflorescence do not open at the same time. As you can see in Fig. 1, there are flowers in different phenological stage in the inflorescence. Therefore, while some petals of some flowers have already gone and pericarpial nectary attracts ants, others are in anthesis or pre-anthesis (Please see a clear example in Fig. 1e where an ant is visiting a pericarpial nectary, close to a opened flower and a flower bud). We have now added this information in the manuscript (L. 65-67)

Results

  1. The title of table 2 is imprecise. The authors call all birds and insects floral visitors, and divide them into two groups EEP and and FV - floral visors using the same name.

Response #9 – thank you for spotting it. We now changed to visitors non-pollinators, and added more information about this classification on lines 291-296.

  1. Figure 2a. The labels of the control and ant group in the legend are unnecessary, because this information is presented below the X axis. The standard error is presented graphically with whiskers usually. What did the authors mean when they wrote “Large symbols represent mean±SE”?

Response #10 – We now removed the legend for color. The error bars are very small and the large symbol representing the mean was overlapping them. We have now updated our figure (please, also check our Response#2) decreasing the large symbols. Large symbols are the larger in the figure showing the mean plus/less the standard error. Legend was now updated.

  1. If I read the legend to Figure 2b correctly, the results do not correspond to the interpretation in subsection 2.2.2. According to Fig. 2b, there are less than 50% on Miconia inflorescences after 300 seconds and on Pollicourea inflorescences after 600 sec. More than 50% of termites stay on Policourea leaves and 100% of termites are present on Miconia leaves at all times. This mistake is also in line 127.

Response #11 – Thank you for spotting it. The colors in the legend were not correct. We have updated our figure matching the results and discussion.

  1. I suggest writing, the percentage of ants remaining as a percentage that survived.

It avoids contradiction when the authors write that less than 50% termites was found by ants and the survival curve presents more than 50%

Response #12 – We now updated our sentence using percentage of survivorship “Vertical and horizontal black lines in (b) represent the time where 50% of the termites survived in each group (more than 50% of termites survived in Miconia)”

  1. I suggest unifying (percentage or decimal value) the presentation of fruit production probability and fruit ratio in the text and in Figure 3.

Response #13 – We now standardized to percentage.

  1. It should be very clearly highlighted that in Experiment 3, the Ants group refers to one ant species and the Control group to other ant species.

Response #14 – We now added this information in the results (L. 145-150)

  1. If the no-ants group and the ant-excluded group (experiment 1 and 4) are similarly constituted, why the different names?

Response #15 – We used different names to make it easier for readers to distinguish between the experiments. Using the same name can be confusing and make it difficult to identify which specific experiment is being referred to

Methods

  1. Line 270-275 This paragraph does not relate directly to the study method and should rather be mentioned in the introduction.

Response #16 – We have included additional information about flower biology and pollination (L. 64-66, L. 79-80) in the introduction, providing background to readers (also see our Response #7). We kindly request that this information be retained in the methods section as well, as it succinctly summarizes the main characteristics of the species under study within a single paragraph, whereas these details are dispersed throughout the introduction.

  1. How many replications of flower observations were made during November-February? Have the plants been blowing flowers all this time?

Response #17 – The entire study was conducted from November to February. However, in each survey or experiment we have a different methodology with different observations and sample size. Experiment 1 – 30 shrubs, 3 leaves per shrub, 2 analyzes of herbivory, one when leaves were young and another when mature (L. 295-307). Experiment 2 – 15 P. rigida, and 15 M. albicans, two termites flowers and two termites leaves (L. 309-316). Experiment 3 – 60 plants, four treatments, 30 min per plant from 6:00 am to 6:00 pm (L. 321-350). Experiment 4 – 26 plants, two groups. Plants checked weekly until production of fruits (L. 356-361).

In our region, plants are usually in bloom from September to March [23]. We now added this information in the text (L. 282-283)

  1. What were the rules for dividing floral visitors into EEP and FV? The frequency of Euglossa bee visits was the same as E. macroura. Why is this bee excluded as a pollinator? Did insects collect nectar from pericarpial nectaries?

Response #18 –To attend the reviewer, in methods we included the following text: We consider effective pollinators (EFPs) as those animals that legitimately harvested nectar by opening the corolla and making contact with the floral structures effectively. Floral visitors that did not pollinate the plants by accessing nectar from the base of the corolla or directly from the pericarpial nectaries without making contact with the plant's reproductive structures were classified as visitors non-pollinators (VPNs).

Euglossa bees do not contact the reproductive structures of these flowers, they collect nectar in the basis of corolla, or they are so small that access the floral tube without contacting reproductive structures or access nectar direct in the pericarpial nectaries. While the hummingbirds introduce its tongue and part of the head in corolla.

Although other insects can visit PNs, most of the PNs visitors are ants.

  1. It is necessary to explain how the probability of fruit production was calculated. What was the meaning of recording and measuring the visitation time of pollinators? And were these variables included in the model of probability of the fruit production?

Response #19 – As we recorded the number of flower buds before counting the number of fruits, this analysis offers the probability of fruit production (whether produced or not) based on the count of flower buds. This variable was considered the response variable in our model. We now added this information in the text (L. 381-385). We now removed the sentence about recording the visiting time of pollinators. Although we record some visits, we did not directly use this data in the manuscript.